# Salt-Mediated Organic Solvent Precipitation for Enhanced Recovery of Peptides Generated by Pepsin Digestion

**DOI:** 10.3390/proteomes9040044

**Published:** 2021-11-03

**Authors:** Venus Baghalabadi, Habib Razmi, Alan Doucette

**Affiliations:** 1Department of Chemistry, Azarbaijan Shahid Madani University, Tabriz P.O. Box 53714-161, Iran; Venus.b@dal.ca (V.B.); h.razmi@azaruniv.ac.ir (H.R.); 2Department of Chemistry, Dalhousie University, Halifax, NS B3H 4R2, Canada

**Keywords:** precipitation, acetone, peptides, pepsin, low molecular weight, sample preparation, mass spectrometry

## Abstract

Conventional solvent-based precipitation makes it challenging to obtain a high recovery of low mass peptides. However, we previously demonstrated that the inclusion of salt ions, specifically ZnSO_4_, together with high concentrations of acetone, maximizes the recovery of peptides generated from trypsin digestion. We herein generalized this protocol to the rapid (5 min) precipitation of pepsin-digested peptides recovered from acidic matrices. The precipitation protocol extended to other organic solvents (acetonitrile), with high recovery from dilute peptide samples permitting preconcentration and purification. Mass spectrometry profiling of pepsin-generated peptides demonstrated that the protocol captured peptides as small as 800 u, although with a preferential bias towards recovering larger and more hydrophobic peptides. The precipitation protocol was applied to rapidly quench, concentrate, and purify pepsin-digested samples ahead of MS. Complex mixtures of yeast and plasma proteome extracts were successfully precipitated following digestion, with over 95% of MS-identified peptides observed in the pellet fraction. The full precipitation workflow—including the digestion step—can be completed in under 10 min, with direct MS analysis of the recovered peptide pellets showing exceptional protein sequence coverage.

## 1. Introduction

Optimal proteome characterization by mass spectrometry is critically dependent on the successful isolation and recovery of high-purity samples ahead of analysis. To that end, protein precipitation has become an indispensable tool for proteome sample preparation. Organic solvent-based precipitation is particularly favored for its capacity to recover intact proteins in high yield, although this generally reveals a bias in terms of reduced recovery of lower molecular weight species [1]. Our group demonstrated the critical role of ionic strength solution to enable the precipitation of proteins in 80% acetone [1,2]. We also concluded that the combination of higher levels of acetone (97%), together with specific salt types (e.g., ZnSO_4_), would facilitate the recovery of low mass peptides resulting from tryptic digestion [3]. In these experiments, test samples were subject to desalting prior to precipitation to eliminate the potential of the solution matrix interfering with the isolation of peptides while disclosing the role of specific salt types and concentrations.

Pepsin is an important acidic protease that is widely applied in the hydrolysis of proteins [4]. It preferentially cleaves at the C-terminus of phenylalanine, leucine, tyrosine, and tryptophan residues. Consequentially, peptic peptides are chemically more diverse than tryptic peptides. Pepsin is used for collagen and gelatin extraction [5,6,7,8]. In proteomics, it is used for the in-depth characterization of antibodies as biopharmaceuticals [9]. Pepsin also plays a critical role in hydrogen/deuterium exchange experiments [10,11], which require a low temperature and low pH to preserve the isotopic labels on peptides for characterization. Its resistance to high-temperature digestion makes pepsin ideal for proteolytically resistant, tightly folded proteins [4,12,13]. When subjecting proteins to pepsin digestion over trypsin, higher sequence coverage is generally provided, which can increase the number of available post-translational modifications for analysis. Proteoforms that share a high degree of sequence homology can also be more easily distinguished this way. To date, few studies have focused on the recovery of pepsin peptides through precipitation. Primarily, these investigations focused on the recovery of collagen peptides by salting out at high ionic strength (>2 M) in acidified solutions [14,15]. A generalized protocol for a preconcentration of pepsin-digested proteins would be useful in MS-based proteome characterization.

Here, we investigated the precipitation of low molecular weight peptides generated by pepsin digestion. We observed that the recovery of these peptides is independent of the acidic solution matrix, though this is critically reliant on the inclusion of specific salts together with an organic solvent. Independently varying the salt and solvent concentrations gives rise to sigmoidal recovery trends, with an optimal yield obtained at above 100 mm salt and 90% solvent. This protocol suggests a convenient approach to quench and recover peptide-digested proteins ahead of MS analysis, providing maximal higher protein sequence coverage when subject to bottom-up MS analysis.

## 2. Materials and Methods

### 2.1. Materials

Standard proteins (bovine serum albumin, carbonic anhydrase and myoglobin, equine cytochrome c, alpha casein, and chick lysozyme), as well as porcine pepsin (*EC 3.4.23.1*, cat# P-7000), were purchased from Millipore Sigma (Oakville, ON, Canada). Bovine plasma (cat # P4639) was obtained as a dried powder from Millipore Sigma and reconstituted in water to a protein concentration of 2 g·L^−1^. A yeast proteome extract was prepared from *S. cerevisiae*, cultured overnight to an OD of 1 by standard protocols. Proteins were extracted by grinding the yeast cell pellet under liquid nitrogen with a mortar and pestle (final concentration, 2 g·L^−1^). The Pierce BCA assay kit used for determination of protein concentration, salts (aluminum chloride, ammonium chloride, ammonium sulfate, calcium chloride, magnesium sulfate, sodium chloride, sodium sulfate, zinc chloride, and zinc sulfate), and organic solvents (acetone and acetonitrile) were from Thermo Fisher Scientific (Ottawa, ON, Canada). Solvents were of HPLC grade, while water was purified to 18.2 MΩ·cm.

### 2.2. Pepsin Digestion

Proteins were digested with pepsin at a concentration of 1 g·L^−1^ using one of two protocols: For optimization of the peptide precipitation protocol, proteins were diluted into HCl (final 0.1 M) and combined with pepsin at a 50:1 mass ratio of protein to pepsin. Digests proceeded overnight at 37 °C and were terminated by heating the solution to 100 °C for 5 min. The standard protein mixture was subject to disulfide bond reduction with 5 mm dithiothreitol (30 min, 56 °C), followed by alkylation with 11 mm iodoacetamide (room temperature, dark, 30 min), prior to digestion. For MS analysis, protein samples were diluted into formic acid (final 1% *v*/*v*), to which pepsin was added at mass ratios of 10:1, 100:1, and 1000:1 (protein:enzyme). Samples of these solutions were incubated at room temperature for 1 and 10 min, after which the digests were terminated by subjecting the complete sample to immediate precipitation through the addition of salt and acetone (Section 2.3). The precipitated peptides were retained following centrifugation and stored in the freezer as a dried pellet for up to 1 week prior to MS analysis (Section 2.5).

### 2.3. Peptide Precipitation

For precipitation, pepsin-digested peptides were combined with one of the various salts over a range of concentrations, as specified in the results section. After briefly mixing the sample, the appropriate volume of organic solvent (acetone or acetonitrile) was added, and the test sample was gently mixed to combine the solvents. The reported salt concentration reflects the value in the sample prior to the addition of organic solvent, while the percentage of organic solvent is that of the total solution. The final volume of the sample, including the organic solvent, was maintained at 1000 µL, while 15 µg of digested peptides were routinely used for precipitation (except when investigating the effects of peptide concentration on precipitation recovery). Samples were incubated at room temperature for 5 min, after which the pellet was isolated by centrifugation (13,000× *g*, 2 min). The retained supernatant was removed with a pipette, with care taken not to disturb the pellet while leaving behind less than 5 µL of solvent (under 0.5% of the total supernatant). The pellet was air dried, while solvent from the supernatant fraction was removed by SpeedVac for subsequent analysis. Peptides from the pellet and supernatant fractions were reconstituted in water, with a brief vortex mixing (1 min), followed by repeated pipetting of the solution in the vial (1 min) to enable quantitative analysis (Section 2.4). For MS analysis (Section 2.5), the dried pellet was reconstituted in a similar fashion, although 5% acetonitrile and water with 0.1% formic acid were used as the resolubilizing solvent. In all instances, the resolubilized sample was centrifuged, retaining the clarified solution as a precaution to remove any portion of the sample that may not have redissolved.

### 2.4. Peptide Quantitation by LC-UV

Resolubilized peptides were injected onto a 50 × 1 mm self-packed HPLC column containing Poros 20 R2 beads (Thermo Fisher Scientific, Oakville, ON, Canada). The Agilent 1200 system operated at a flow rate of 100 µL/min. The gradient comprised a stepwise increase from 5% to 85% acetonitrile in water, 0.1% TFA, 5 min after sample injection. Peptides eluted as a single peak, and the resulting area, monitored by UV absorbance (214 nm), was compared to a calibration curve consisting of control samples of pepsin-digested BSA that did not undergo the precipitation workup. The eluting peptide peak was retained via a fraction collector. Solvent from these fractions was removed in a Speedvac before peptides were resolubilized as described (Section 2.3) ahead of MS analysis.

### 2.5. LC-MS/MS Analysis

For BSA and the standard protein mixture, peptides precipitated using optimized conditions (100 mm ZnSO_4_, 97% acetone, and incubated at room temperature for 5 min) were resolubilized (Section 2.3) and directly subjected to bottom-up LC-MS/MS analysis to characterize peptides. The equivalent of two picomoles per protein was injected onto a 75 μm × 30 cm column (3 μm C18 Jupiter beads, Phenomenex, Torrance, CA, USA), self-packed within a PicoFrit nanospray emitter (New Objectives, Littleton, MA, USA) and interfaced to an LTQ linear ion trap mass spectrometer (Thermo Fisher, San Jose, CA, USA). An Agilent 1200 HPLC system was used to deliver a gradient from solvent A (water, 0.1% formic acid) to solvent B (acetonitrile, 0.1% formic acid) at a flow rate of 0.25 μL/min. The gradient was initially set to 5% B and held for 5 min. The organic solvent was increased to 35% by 70 min, 95% by 80 min, and lowered to 5% at 81 min. MS was operated in data-dependent mode, which cycles from a full MS scan to a zoom scan for charge state determination, followed by MS/MS of the top five ions with a collision energy of 35. Charge-state screening was enabled to ignore singly charged ions, ions with a charge of 4 and greater, or ions where the charge state could not be assigned. The mass range was from 400−1300 m/z.

For analysis of the alpha casein digest, as well as the yeast and plasma proteome mixtures, peptides were subject to LC cleanup (Section 2.4), before being analyzed on an Orbitrap Velos Pro (Thermo Fisher Scientific, San Jose, CA, USA) connected to a Dionex Ultimate 3000 LC nanosystem (Thermo Fisher Scientific, Bannockburn, IL, USA) with a self-packed monolithic C18 column and a 10 μm PicoTip noncoated Emitter Tip (New Objective, Woburn, MA, USA). The linear LC gradient increased from 0.1% formic acid in water to 35% acetonitrile over 2 h. MS operated in data-dependent mode at a resolution of 30,000 full width at half-maximum (FWHM) for MS^1^, scanning in rapid mode for MS^2^ (66,666 Da·s^−1^), with a resolution of <0.6 Da FWHM.

### 2.6. Data Analysis

Peptide MS/MS spectra were searched using the Proteome Discoverer software, version 1.4 (Thermo Fisher Scientific, San Jose, CA, USA). Standard proteins were searched against a FASTA sequence file compiled for these proteins (plus pepsin). The *S. cerevisiae* FASTA database contained 9931 entries (downloaded 29 May 2019). With bovine plasma, a FASTA database was assembled from the top 20 most abundant proteins. The enzyme was set with unrestricted cleavage. A mass tolerance of 1.5 Da (MS) and 0.8 Da (MS/MS) were used for LTQ data, while 10 ppm (MS) and 0.8 Da (MS/MS) were employed with the Orbitrap Velos Pro. Scoring parameters were optimized to provide a peptide false discovery rate of 1% or less. The identified peptides were processed using MS Excel and online software to create Venn diagrams [16] and calculate the grand average of hydropathicity (GRAVY) scores [17], along with peptide isoelectric points (pI) [18].

Peptides identified from the pellet vs. supernatant fractions were statistically compared in terms of various properties (MW, length, pI, GRAVY score) using pairwise t-testing. Additionally, the relative frequency of individual amino acids on peptides in the respective pellet and supernatant fractions was compared using pairwise t-testing (two-tailed in all cases). Calculated *p* values are listed in the manuscript and denoted by asterisks in the figures.

## 3. Results

### 3.1. Salt Controls the Precipitation of Peptic Peptides in Organic Solvents

When considering intact, large molecular weight proteins (>10 kDa), maintaining a sufficient ionic strength (typically from 100 mm NaCl) along with elevated levels of acetone (80%) is a critical factor to enable high-efficiency precipitation of essentially all protein types [1,2]. Intact protein recovery exceeds 90% when precipitated with NaCl, though these larger components remain fully soluble in 80% acetone when the solvent is added below a critical solution ionic strength [3]. However, when we attempted the combination of 100 mm NaCl in acetone to recover low molecular weight peptides generated from the pepsin digestion of BSA, we did not observe a statistically significant increase in precipitation efficiency relative to the ‘no salt’ control (Figure 1A). The majority of these peptides have a molecular weight below 2 kDa. An analysis of the distribution of pepsin-generated peptides recovered in the pellet is provided in Section 3.4. Increasing the concentration of NaCl beyond 100 mm did not significantly alter recovery. By sharp contrast, the inclusion of different salt types, namely ZnSO_4_ (100 mm) and, to a lesser extent, MgSO_4_, with acetone, provided a vast improvement in peptide recovery. With ZnSO_4_, over 60% of the total peptide content by mass was obtained in the pellet. Interestingly, neither ZnCl_2_ nor Na_2_SO_4_ could facilitate this high level of recovery. This provides a clear indication that neither Zn^2+^ nor SO_4_^2−^ are solely responsible for enabling the desired level of peptide aggregation. None of the chloride salts tested facilitated a precipitation recovery that was statistically higher than the no-salt control. In addition, while sulfate ion showed a favorable recovery when paired with either zinc (II) or magnesium (II) cations, the inclusion of ammonium sulfate—being the preferred kosmotopic ion of the Hofmeister series for salting out compounds from aqueous solution [19]—did not show improved recovery when attempts to precipitate peptides in an organic solvent were made. Regarding the acidic pH employed to precipitate the pepsin-digested proteins, we suspect that cations would play a more significant role in pairing with electron-donating amino acid residues. However, the different yields observed for chloride vs. sulfate salts demonstrate that both the cation and anions present in the sample solution must play important roles in facilitating peptide aggregation. These observations contrast significantly with the conventional phenomenon of salting out, whereby anions of the Hofmeister series tend to play a more significant role [19].

Within the single-phase mixed aqueous-organic solvent system, the salt concentration required to enable precipitation of pepsin peptides is significantly lower than that typically needed to salt out proteins from purely aqueous systems. As seen in Figure 1B, a sigmoidal-shaped peptide recovery curve is generated when varying the concentration of ZnSO_4_ in a solution with 70% acetone. Peptide recovery plateaus at approximately 200 mm salt, although an ED50 of only ~5 mm salt is observed, sufficient to recover half of the observed maximum yield. Salt concentrations below 0.1 mm are indistinguishable from the no-salt control, with a non-zero recovery presumably obtainable owing to ions inherent to the sample. Similar sigmoidal curves have been reported for intact proteins, although with a steeper rise in recovery spanning only one order of magnitude [1,2].

### 3.2. Higher Organic Solvent Is Needed to Precipitate Peptides

With larger molecular weight proteins, the use of 80% acetone is sufficient to maximize recovery through precipitation, with no additional gains observed with a higher percentage of organic solvent. However, at the peptide level, increasing the organic solvent concentration beyond 80% can lead to higher sample recovery through precipitation. For peptides generated from trypsin digestion, we previously reported a linear correlation between recovery and acetone concentration for these peptides at solvent levels above 60% [3] but, as seen in Figure 2A, the recovery of pepsin peptides follows a sigmoidal curve with respect to solvent concentration. Moreover, while no recovery of trypsin-digested BSA was previously observed from a solution of 60% acetone, the precipitation of pepsin peptides yielded a recovery of above 50% from this level of acetone. Furthermore, although additional gains were observed as the acetone was increased from 85% to 97%, the recovery increase was no longer linear, as seen with trypsin peptides. Nonetheless, at the peptide level, it is apparent that higher levels of organic solvent result in an increase in precipitation yield.

Precipitation of low molecular weight pepsin peptides through the combination of salt and organic solvent extends beyond the use of acetone. As seen in Figure 2B, similar peptide recovery is observed when substituting the organic solvent for acetonitrile. This solvent has previously been employed to recover low molecular weight plasma peptides [20], though not in combination with a particular salt. The two solvents tested in Figure 2B provide similar recovery and reproducibility over the range of concentrations employed.

### 3.3. Precipitation of Dilute Peptide Solutions

The recommended protocol for the precipitation of pepsin-digested peptides includes 100 mm ZnSO_4_ with 97% acetone (5 min under room temperature). This amount of organic solvent is achieved by combining 15 µL of the initial peptide solution, 15 µL of salt (200 mm), and 970 µL of acetone. One might suspect that this level of dilution of the original sample with the organic solvent could compromise recovery. The recovery data of Figure 2 demonstrates that this is not the case. Furthermore, the precipitation protocol can be applied to recover peptides from dilute solutions. As shown in Figure 3, a 50-fold drop in the initial peptide concentration (from 6.5 g/L to 0.1 g/L) decreases the peptide recovery (t-stat 26.9 > t-critic 4.3/*p* value = 0.0014). However, total yields are above 70%, which justifies applying the precipitation protocol to more dilute samples.

### 3.4. Mass Spectrometry of Precipitated Peptides

When subjected to the optimized precipitation protocol, the total recovery of pepsin-generated peptides from the pellet is above 90% (Figure 4A). These low molecular weight peptides are readily resolubilized in a conventional LC-MS solvent (0.1% formic acid in water with 5% acetonitrile) with minimal mixing. A reversed-phase capture and analysis of peptides in the supernatant resulted in a retention of the residual 10% of the sample that did not precipitate (Figure 4A), suggesting that the unprecipitated fraction primarily comprises peptides—as opposed to free amino acids—which would not be retained by the reversed-phase column. It was of interest to note if the recovered peptides were representative of the complete distribution of sample components or if there were statistical differences in the composition of peptides, leading to differences in precipitation efficiency. Thus, each fraction (pellet, supernatant) was subject to bottom-up LC-MS/MS analysis to profile the composition of peptides within the respective samples. The base peak chromatograms shown in Figure 4B,C represent equivalent mass loading (2 pmol) of each sample (a pepsin-digested mixture of five standard proteins), meaning that the supernatant fraction was concentrated approximately 10-fold, relative to the pellet. From Figure 4, clear differences are observed in the peptide composition between the two fractions, suggesting preferential precipitation of specific peptides.

The distinct profiles shown in Figure 4 are representative of a mixture of pepsin-digested protein standards. While the diversity of peptides present in such a sample is significant, the total number of peptides remains small relative to a complex biological proteomic mixture. The optimized peptide precipitation protocol was therefore applied to the pepsin digests of two complex systems: a yeast whole-cell proteome extract, and a sample of bovine plasma. Notably, the MS system available for this study, an Orbitrap Velos Pro, is a decade-old platform that limits the capacity for proteome profiling. However, it is not our goal to generate a comprehensive list of protein identifications but rather to disclose if there are any relative differences in peptides recovered from the supernatant and pellet fractions. This provides a measure of the overall applicability and the potential limitations of our low molecular weight peptide precipitation protocol.

Bottom-up MS analysis of peptides recovered in the pellet and supernatant fractions resulted in the cumulative identification of 4049 unique peptides, with 3813 in the precipitation pellet and 236 observed in the supernatant fractions. The diversity of components from this list has afforded us the opportunity to uncover trends in the relative precipitation efficiency of peptides. The complete listings of MS-identified pepsin peptides are presented as Appendix A. The Venn diagrams of Figure 5 summarize the total peptide composition resulting from an analysis of the yeast and plasma samples (a comparison of protein identification is provided in the Appendix A). From these numbers, it is apparent that over 93% of the identified proteins were observed in the pellet fraction. Of the peptides observed in the supernatant, approximately 50% were also detected in the pellet fraction. Less than 4% of identified peptides were unique to the supernatant fraction. These peptides may have also partitioned to a certain extent into the pellet fraction. Nonetheless, we examined the distribution of peptides that were preferentially observed in the supernatant to uncover the characteristic properties favoring precipitation.

Figure 6A examines the relative abundance of peptides recovered in the pellet and supernatant fractions. For peptides that were commonly identified across the two fractions, we observed a higher number of peptide spectral matches (PSM) in the pellet fraction compared to the supernatant (median PSM ratio 1.5)—indicating that the commonly identified peptides were generally more abundant in the pellet fraction. We next considered the cumulative distribution of unique peptides recovered in the pellet vs. supernatant fractions from the precipitation of yeast, plasma, and standard proteins, with respect to their molecular weight, hydrophobicity (GRAVY score), and isoelectric point. These results are plotted in Figure 6B,D, respectively. A clear difference emerged with respect to the molecular weight when examining for peptide properties that lead to more favorable precipitation. Comparing uniquely identified peptides, the pellet displayed a median MW of 1616 u, vs. 1125 u in the supernatant, clearly indicating that smaller peptides are more difficult to precipitate (*p* value of 6 × 10^−58^ from pairwise t-test). Nonetheless, 803 (20%) of the peptides identified in the pellet had a molecular weight below 1000 u. The smallest pepsin-generated peptide observed in the precipitation pellet consisted of six amino acids with a mass of 559 u. Similarly, noted statistical differences were observed in the peptide hydrophobicity (average GRAVY score of 0.05 in the pellet and 0.6 in the supernatant) and isoelectric point (mean pI 6.7 in the pellet vs. 5.5 in the supernatant). Thus, peptides with the most favorable recovery through the optimized precipitation protocol constitute a higher molecular weight, higher polarity (lower GRAVY score), and a higher isoelectric point. However, it is critical to note that peptides at the extremes in each of these categories were still recovered in the pellet fraction. In other words, smaller, less polar, and low pI peptides can still be precipitated with the optimized protocol, though with lesser efficiency than the average 90% yield observed across all peptides.

Beyond these general peptide properties, we also examined the amino acid makeup of the precipitated peptides to determine if specific residues correlate to a higher yield. A prior examination of tryptic peptides [3] revealed that 95% of all peptides contained at least one acidic residue (100% of tryptic peptides contain a basic residue). From the present study, it was evident that certain amino acid residues are statistically over or under-represented in the pellet peptides compared to those of the supernatant (Table 1). The greatest statistical difference was in the basic residues, histidine, lysine, and arginine, which were each overrepresented in the pellet peptides.

Grouping the amino acid residues according to structural class also highlights a strong correlation, wherein precipitated peptides contain a higher proportion of aliphatic residues (A, G, I, L, P, and V; see Appendix A). This is in agreement with the global hydrophobicity of peptides collected in the pellet being statistically greater (i.e., a higher negative GRAVY score) than those of the supernatant. Despite this, examination of pepsin-generated peptides revealed that no single amino acid residue nor residue class (acidic, basic, polar, aromatic, or sulfur-containing) was deemed essential to induce precipitation of the peptide (see Appendix A). For example, the hypothesis that peptides must contain a histidine residue to bind with zinc is disproven, as peptides were recovered in a pellet that did no contain histidine. The same is apparent for all individual residues. Thus, precipitation does not rely on any specific residue or class of residues. Rather, salt-mediated solvent precipitation is generally favored for larger, more hydrophilic peptides.

### 3.5. High Sequence Coverage for Pepsin-Digested Proteins

The optimized precipitation protocol reported here provides a convenient means to recover peptides generated from pepsin digestion. Adding a high organic solvent content quenched enzyme activity, allowing for rapid isolation of unique digestion products from pepsin digestions conducted under differing conditions. Table 2 summarizes the peptides identified from the precipitation pellet of pepsin-digested BSA. The complete peptide lists are provided in Appendix A, while Appendix A summarize the data. Employed were six different digestion protocols, ranging in time (1 vs. 10 min) and enzyme ratio (1:10, 1:100, and 1:1000). Since faster digestion, or digestion conducted with fewer enzymes, should result in a greater number of missed cleavage sites, this would generate a greater variety of peptides and enhance sequence coverage. As seen from the table, no individual pepsin digestion condition provided greater than 63% coverage (from 10 min digest, with 1:10 ratio of enzyme: protein). However, the aggregated data provided 78.3% sequence coverage for BSA, with 196 uniquely identified peptides observed in the precipitation pellets (see Appendix A). Venn diagrams comparing the peptides from each of the digestion conditions are also provided as Appendix A. Given a precipitation time of 5 min, with an added 1 min required to centrifuge and 2 min to resolubilize, the complete sample workup can be complete in under 10 min when paired with a 1 min pepsin digestion.

The rapid pepsin digestion and precipitation protocol were also applied to a sample of alpha casein, which contains multiple phosphorylation sites. Only the 10 min digestion was employed. Of the 486 unique mass features corresponding to casein peptides (including alpha S1, S2, and kappa casein), 152 (31%) constituted phosphorylated peptides (Appendix A). From these, all but one was recovered in the pellet fraction. Over 96% sequence coverage was obtained from the alpha S1 casein peptides recovered in the pellet. The region spanning the missing coverage (residue 78–84, ES*ISS*S*EE) was not observed in the supernatant. This acidic and triply phosphorylated peptide may not have been retained on the LC column.

### 3.6. Reproducibility of Peptide Precipitation

As with any sample preparation strategy, the optimized precipitation protocol may introduce unwanted sample variability. While the characteristics of individual peptides (e.g., lower MW, more hydrophobic) may lend a lower precipitation yield, the high distribution of peptides identified in the pellet fraction (approaching 95%) indicates that the majority of peptides will be successfully precipitated. Perhaps more importantly, we examined the reproducibility of precipitating independent samples of pepsin-digested yeast, relative to the repeatability of technical MS replicates of an identical sample. As seen in Figure 7A,B, the degree of overlap between the replicates was equivalent across each set of samples. This indicates that the precipitation step adds minimally to the overall variability in the proteomics workflow. Moreover, a quantitative comparison of the proteins identified from the various replicates shows an exceptional overlap, as indicated by the slope (approaching one) when plotting the protein PSMs from the MS replicates (Figure 7C), as well as the precipitation replicates (Figure 7D).

## 4. Discussion

The precipitation of low molecular weight peptides in organic solvents is facilitated by the inclusion of specific salts in the sample at sufficiently high ionic strengths. When combined with high levels of organic solvent, peptide recovery of over 90% is possible from a pepsin-digested protein sample. The most favorable salt tested thus far, ZnSO_4_, is not impacted by the inclusion of an acidified matrix (either 0.1 M HCl or 1% formic acid), in that the presence of specific cation and anion species will influence peptide recovery. While higher levels of organic solvent improve recovery, diminishing gains are realized above 85% solvent, which contrasts with the linear relation between solvent content and recovery observed for tryptic peptides.

A distinction between tryptic and pepsin-digested peptides relates to the absence of a basic residue at the C terminal end of pepsin peptides. From the current study, the presence of a positively charged basic residue (inherent to tryptic peptides) is not essential to induce peptide precipitation (see Appendix A). Zn^2+^ is a preferred ion to maximize precipitation efficiency, despite employing an acidified sample matrix, which would reduce the concentration of negative charge residues on the peptides. As a Lewis acid, Zn^2+^ (as well as Mg^2+^) can interact with electron-donating species, which includes the amide backbone of the peptide [21]. This justifies why the presence of specific amino acid residues, such as histidine or cysteine, is not required on the peptide to induce precipitation. In fact, no single amino acid nor class of amino acids (including acidic, basic, polar neutral, aromatic, or sulfur-containing residues) is required to mediate peptide precipitation (Appendix A). The presence of charged residues, including the N or C terminus of a peptide, contributes significantly to its solubility in an aqueous solvent. To induce precipitation, these charges should be minimized. The enhanced ion pairing between salt ions and peptides, as is observed in a solvent containing a high concentration of organic solvent, is therefore presented as a possible explanation for peptide precipitation. This is supported by the increased prevalence of basic (positively charged) amino acid residues in the pellet relative to peptides identified from the supernatant fraction. An examination of the mechanism governing salt-mediated peptide precipitation in organic solvents is ongoing.

The practical utility of this precipitation protocol is realized when profiling pepsin-digested proteins by bottom-up MS. High protein sequence coverage can be obtained when analyzing peptides generated from protein standards, particularly when obtained under different digestion conditions. We demonstrated that the precipitation protocol was applicable to peptides of all amino acid compositions with extremes in molecular weight, polarity, and charge. In the case of alpha-casein peptides, the only segment of the protein ‘missing’ from the precipitation pellet was also not observed in the supernatant and may therefore be unretained or poorly ionized in the LC-MS/MS platform. Similarly, high sequence coverage is also possible from the analysis of proteins in complex mixtures. For example, the immune glycoprotein, complement C3, was identified in the plasma mixture through 119 unique peptides, representing 48% sequence coverage. While a small fraction of peptides (<5%) were uniquely identified in the supernatant fraction, a detailed MS analysis of complex proteomic systems reveals that the optimized precipitation protocol is applicable to all classes of peptides observable in a typical LC-MS/MS experiment. Moreover, quantitative analysis of the recovered peptide fractions is possible, given both the high recovery, combined with exceptional reproducibility of the precipitation protocol. Bottom-up MS analysis introduces variability in terms of identified peptides, as well as peptide abundance, which justifies the use of replicate MS injections to profile the sample. Likewise, inconsistencies during sample preparation may challenge quantitative analysis. However, as we show here, the variation in sample preparation is minimal, particularly relative to the inherent MS variability. To ensure minimal bias, it is recommended to independently precipitate samples in triplicate and pool the resulting pellets into a single sample prior to MS analysis. The precipitation protocol is a convenient means to preconcentrate, purify, and preserve the peptide mixture ahead of MS. Future work is therefore being directed at the analysis of deuterated peptides generated from pepsin digestion in an HDX labelling experiment.

## 5. Conclusions

The combination of specific salts, together with organic solvents, results in high-recovery precipitation of peptides generated from pepsin digestion. The digestion and precipitation workflow is important for proteome analysis by mass spectrometry.

## Figures and Tables

**Figure 1 proteomes-09-00044-f001:**
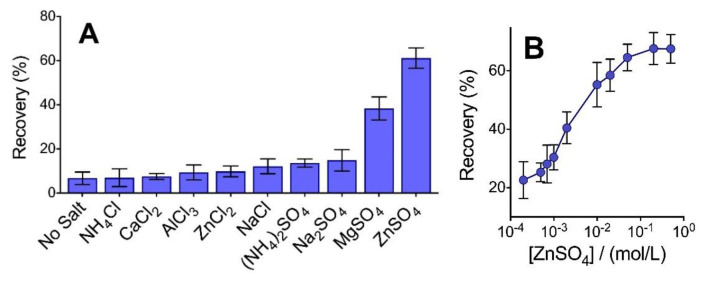
(**A**) The effect of salt type (100 mM) prior to the addition of 70% acetone to the precipitation recovery of BSA peptides generated from pepsin digestion. (**B**) Varying the salt concentration results in a sigmoidal-shaped recovery curve. Error bars represent standard deviation from the independent precipitation of samples in triplicate.

**Figure 2 proteomes-09-00044-f002:**
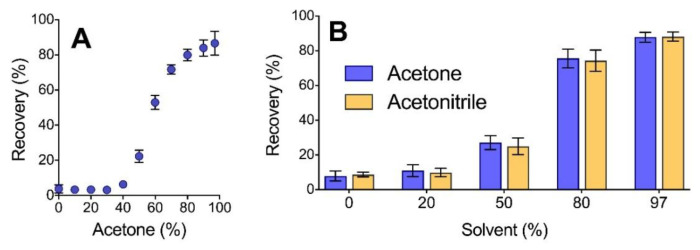
Effect of solvent concentration (**A**) and solvent type (**B**) on the precipitation efficiency of peptides generated from the pepsin digestion of BSA. Samples were precipitated with inclusion of 100 mm ZnSO_4_ prior to addition of acetone. Error bars represent standard deviation from independent precipitation of samples in triplicate.

**Figure 3 proteomes-09-00044-f003:**
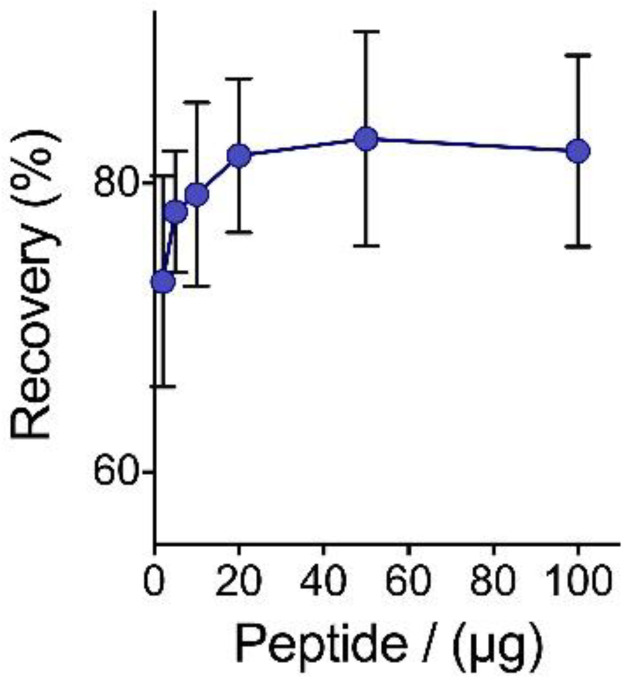
Effect of initial peptide concentration on recovery through precipitation. Sample comprises pepsin-digested BSA, precipitated with 100 mm ZnSO_4_ in 70% acetone. Error bars represent standard deviation from 3 independently precipitated samples.

**Figure 4 proteomes-09-00044-f004:**
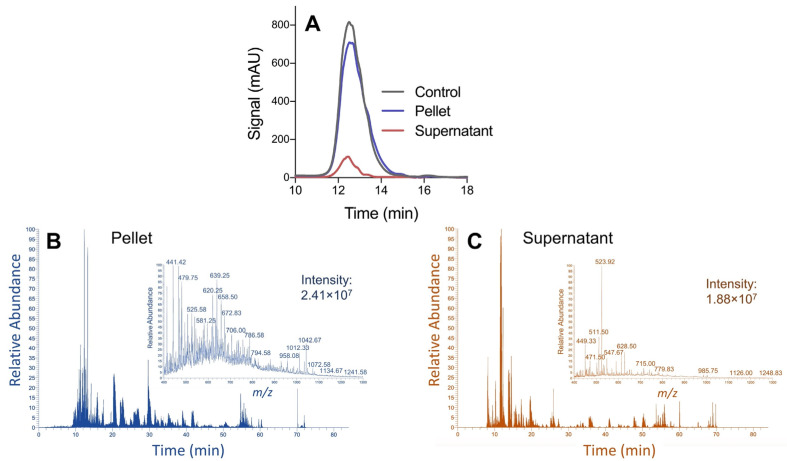
(**A**) Quantitative analysis of peptides recovered in the precipitation pellet and supernatant fractions through reversed-phase LC, coupled with UV absorbance. (**B**,**C**) The resulting base peak chromatograms from LC-MS/MS analysis of the fractions demonstrate clear differences between samples. The inset shows the cumulative MS spectra from each sample, suggesting greater complexity in the pellet. The sample comprises a mixture of 5 standard proteins, following digestion with pepsin.

**Figure 5 proteomes-09-00044-f005:**
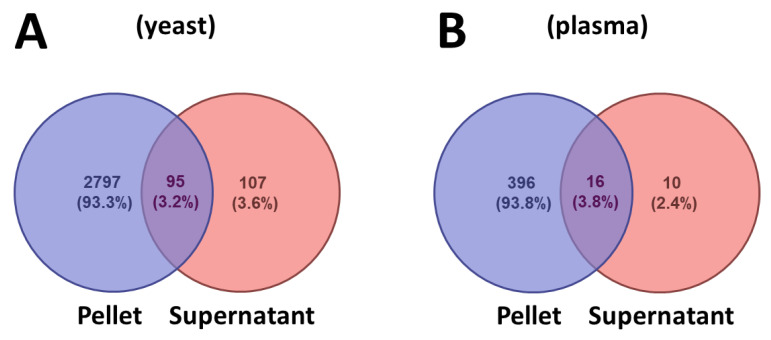
Venn diagram summarizing the peptides identified from bottom-up MS analysis of the pellet and supernatant fractions of (**A**) pepsin-digested yeast and (**B**) digested bovine plasma, following precipitation.

**Figure 6 proteomes-09-00044-f006:**
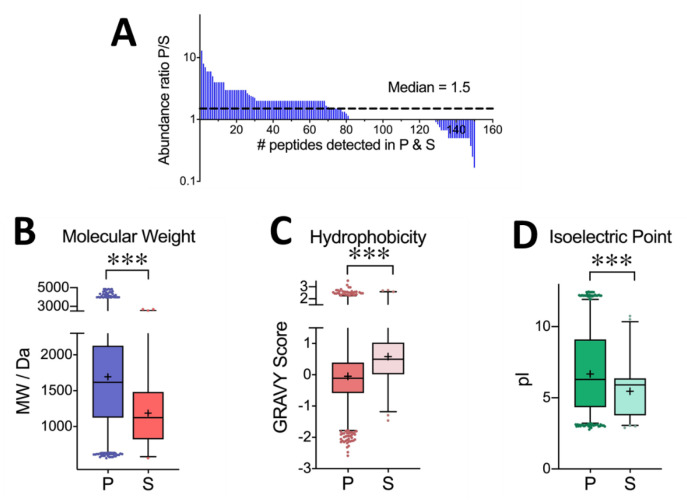
(**A**) Relative abundance of peptides commonly observed in the pellet (P) and supernatant (S) according to a ratio of peptide spectral matches (PSM). The characteristic peptide properties of each fraction are compared with respect to: (**B**) peptide molecular weight (**C**); hydrophobicity, represented by a calculated GRAVY score; and (**D**) isoelectric point. Pairwise t-testing indicates statistical differences across each of these three properties (*p* < 0.001). Statistically significant differences are represented by asterisks in the figure, where *** denotes *p* < 0.001.

**Figure 7 proteomes-09-00044-f007:**
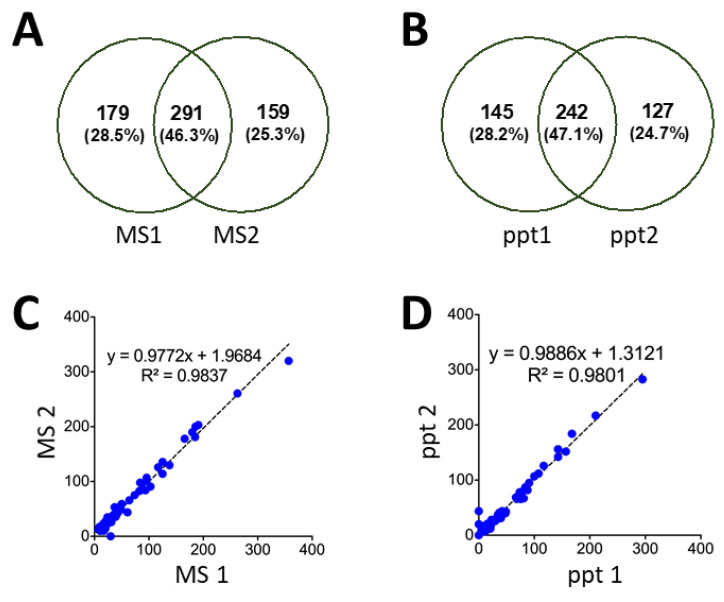
Venn diagram showing overlapping peptides with PSM >3, identified in the precipitation pellet from pepsin-digested yeast. (**A**) Consists of replicate MS analysis of a single sample, while (**B**) compares two independently precipitated samples of yeast. Quantitative analysis of the identified proteins is shown through PSMs per protein, plotting the correlation between (**C**) two MS replicates and (**D**) independent precipitations.

**Table 1 proteomes-09-00044-t001:** Normalized frequency of amino acid residues contained on peptides recovered in the pellet and supernatant fraction following precipitation.

Amino Acid Residues	Average of Normalized ^1^ Residues to the Length of the Peptides)		*p* Value of *t*-Test
	Pellet	Supernatant	Pellet vs. Supernatant
Basic Residues (K + R + H)	0.12 ± 0.08	0.06 ± 0.07	1.46 × 10^−34^
Histidine (H)	0.02 ± 0.03	0.01 ± 0.02	7.84 × 10^−22^
Lysine (K)	0.06 ± 0.06	0.03 ± 0.05	5.96 × 10^−18^
Arginine (R)	0.04 ± 0.05	0.02 ± 0.04	7.31 × 10^−15^
Leucine (L)	0.10 ± 0.09	0.14 ± 0.11	2.40 × 10^−15^
Phenylalanine (F)	0.04 ± 0.06	0.05 ± 0.07	5.32 × 10^−3^
Cysteine (C)	0.00 ± 0.02	0.00 ± 0.01	5.69 × 10^−3^
Aspartic Acid (D)	0.05 ± 0.06	0.05 ± 0.07	2.50 × 10^−2^
Serine (S)	0.06 ± 0.07	0.05 ± 0.07	3.34 × 10^−2^
Tyrosine (Y)	0.03 ± 0.05	0.04 ± 0.07	4.30 × 10^−2^
Isoleucine (I)	0.06 ± 0.07	0.07 ± 0.09	4.52 × 10^−2^
Acidic Residues (D+E)	0.11 ± 0.09	0.11 ± 0.09	ND
Alanine (A)	0.09 ± 0.09	0.09 ± 0.09	ND
Glutamic Acid (E)	0.06 ± 0.07	0.05 ± 0.07	ND
Tryptophane (W)	0.01 ± 0.03	0.01 ± 0.02	ND
Glutamine (Q)	0.03 ± 0.05	0.03 ± 0.05	ND
Glycine (G)	0.08 ± 0.08	0.07 ± 0.08	ND
Proline (P)	0.06 ± 0.06	0.06 ± 0.07	ND
Valine (V)	0.08 ± 0.08	0.08 ± 0.09	ND
Methionine (M)	0.01 ± 0.03	0.01 ± 0.04	ND
Threonine (T)	0.05 ± 0.06	0.05 ± 0.07	ND
Asparagine (N)	0.04 ± 0.06	0.04 ± 0.06	ND

^1^ Normalize residue numbers are obtained by dividing the given residue count by the respective peptide length.

**Table 2 proteomes-09-00044-t002:** Summary of precipitated BSA peptides identified by MS following digestion with pepsin for varying times and enzyme ratios.

Protein/Enzyme Ratio	Time (min)	# Peptide	# PSMs ^1^	% Coverage	Avg Seq Length	Avg MW (u)
10:1	1	89	274	52.6%	14.3 ± 5.5	1695 ± 639
	10	133	440	62.8%	13.8 ± 5.1	1634 ± 607
Total		157	714	64.5%	14.0 ± 5.3	1657 ± 628
100:1	1	83	200	55.2%	14.7 ± 5.2	1722 ± 623
	10	87	210	49.1%	14.10 ± 4.7	1666 ± 565
Total		116	410	57.9%	14.6 ± 5.0	1714 ± 595
1000:1	1	48	111	31.8%	14.4 ± 5.0	1677 ± 591
	10	53	104	47.2%	14.3 ± 5.2	1658 ± 625
Total		73	215	523%	14.8 ± 5.4	1710 ± 636
Aggregated Total		196	1339	78.3%	14.6 ± 5.4	1713 ± 629

^1^ Each sample was subject to duplicate LC-MS/MS analysis of the recovered following precipitation with 100 mM ZnSO4 plus 97% acetone (5 min, room temperature).

## Data Availability

The data presented in this study are available in the Appendix A.

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
