# Peer review of "Salt-Mediated Organic Solvent Precipitation for Enhanced Recovery of Peptides Generated by Pepsin Digestion"

_proteomes, 2021, doi:10.3390/proteomes9040044_

Round 1

Reviewer 1 Report

The manuscript by A. Doucette et al. describes an approach to maximizes the recovery of peptides generated from pepsin-digested protein samples. The proposed approach uses a combination of specific salts together with organic solvent for isolating peptides from complex mixture. The overall results show that high recovery of peptides can be obtained in short time.

The precipitation protocol reported in this manuscript can have great significance in the proteomics field and represents a promising approach in proteome analysis by mass spectrometry.

In my opinion, the manuscript is well organized. the analysis is properly carried out, and the results are clearly described.

One minor point:
To verify the general applicability of the precipitation protocol, it would be interesting to apply the proposed method to more complex samples such as yeast extracts containing a variety of compounds with different properties (MW, pI, hydrophobicity) and compare the results with those obtained with model proteins. 

I recommend publication on Proteomes.

Reviewer 2 Report

Overall the manuscript is well presented and well supported by all included evidence. 

Although the utility of such a method is narrow in scope. the work has been well thought out. clearly presented and would be of use to those working in proteomics. 

While model proteins are shown for the purposes of assessing the utility of the salt based precipitation method, the question remains whether this would work with a complex samples such as a cell digest or tissue sample. 

Although what is presented meets the criteria for publication. the mechanisms presented are still theoretical.  I would encourage the authors to consider trialing this method with a model biological organism to further increase the appeal and truly demonstrate the utility of the method as described. 

If this method is applicable to whole cell lysates then what is presented is essentially a 10 minute digest protocol that can give up to 65% sequence coverage for abundant proteins while enabling sequencing of peptides at the lower limit of detection. 

This experiment needs to be done.  

Reviewer 3 Report

I have read with great interest the manuscript entitled “Salt-mediated organic solvent precipitation for enhanced recovery of peptides by pepsin digestion” by Baghalabadi et al.

While this is clearly an area of interest, specifically for fast scans using the bottom-up approach, the content of the manuscript would benefit considerably if the authors would address the following comments and concerns.

Lines 28-30. Your rationale for precipitation is its ability to, “recover intact proteins in high yield, though generally reveals a bias in terms of reduced recovery of lower molecular weight species.” How is this relevant to your work when you are focusing on peptic peptides?

Lines 48-53. There is a mention of proteoforms and proteome characterization (please briefly describe their critical importance for reader) but only one (or more?) highly purified protein standards were assessed. Why were no native samples analysed (i.e., mammalian tissue or plants)?

Section 2.2. Why use two different methods to prepare peptides? Would it not be best to use a single protocol that mimics the handling of native samples?

Is this protocol specific to pepsin peptides and highly purified BSA (and apparently other purified proteins)? Was in-solution peptic and trypsin digestion tested in parallel and the precipitation results with this protocol then compared?

How was the 5 min incubation time at RT decided upon? What appear to be the initial test experiments looked at 1 and 10 min. Was 5 min then simply chosen as a happy medium? What about temperature?

In this reviewer’s experience, fully solubilizing such pellets of organically precipitated protein (particularly native mixtures) is notoriously difficult to do quantitatively. How was full solubilization of the pellet established? Was there still very fine precipitate present in the suspension?

Does the use of zinc salts result in selective precipitation of phosphorylated peptides, or are they present in the supernatant? Then again, the acidified matrix would likely randomly and non-quantitatively modify certain phosphorylated species such that they will not be identified, even if they were to ionize well in the first place.

Lines 184-185. Repetitive use of “peptides generated from trypsin digestion” and “tryptic peptides” in the same sentence.

Line 192 and Fig 2. The text says more variable recovery of (BSA) peptides at higher levels of acetone, yet the bar graph (Fig 2B) shows substantially smaller error bars with 97% acetone than 80% (which, again, is reversed in the data presented in Fig 2A)?

Fig 3. Considering n=3 independent determinations, despite the statistics presented, the variability is quite pronounced raising questions of quantitative evaluation when comparing two or more different native samples. In this regard, what is the rationale for considering ~70% (or even 90%) recovery sufficient when only a single highly purified protein was analysed? Is this only about identifying amino acid sequences from databases or about quantitative proteoform analysis?

Fig 4 and 5. From the standpoint of the importance of being thorough, these data suggest the importance of analyzing both the pellet and the supernatant fraction. Again, this also raises the issue of quantitative comparisons between complex native samples if only the pellet were to be analysed. I would be curious as to how many false determinations of supposedly significant differences in protein abundance between conditions would arise. Similarly, considering Fig S1, how large is the variability in precipitating large vs small peptides, and how does this translate into identifying proteins from databases depending on the distribution of peptides recovered?

Lines 255-257. Is the difference in MW of unique peptides identified in the pellet vs supernatant significant?

Lines 261-277. While there is no clear influence of amino acids in terms of precipitation, it does seem hydrophobic peptides are better precipitated. It would seem this is perhaps a more important difference in applying the current proposed protocol. It would thus be important to confirm this using an isolate of membrane proteins. Of course, the issue still exists that hydrophobic peptides are handled less effectively in the mass spectrometer. However, the authors also note that this precipitation of hydrophobic peptides is only of ‘minor statistical significance’ (Fig S1) leading one to wonder whether there would be any difference to speak of if n were larger or, more importantly, a native sample were analyzed? Additionally, what do you suggest to anyone wanting to use this protocol? The purpose of precipitation is to isolate all peptides from contaminants but the data shows that 21.3% are not being precipitated. Do you suggest that scientists will also have to do MS on the supernatant? Doesn’t that negate the purpose of precipitation? It is also somewhat irrelevant in Fig S1 to note how much the supernatant was concentrated before analysis considering how concentrated the pellet must be relative to the original suspension.

Lines 303-305. Only true in the case of the highly purified single proteins tested here, Although, apparently more than one purified protein standard was tested? The validity of the statement has not yet been demonstrated for a range of proteoforms in native samples.

Majority of data in graphs and all Suppl data is from the BSA protein standard. Six different purified protein standards were apparently tested? Please provide these data as well. Data concerning application of the protocol to native samples would thus also substantially benefit the interpretation of the results, which currently presume to generalize to complex proteoforms more broadly (see below).

Oxidation and carbamidomethyl modifications are identified in Suppl data, why these specific ones?

Suppl Table 2 (indeed the data in every table) certainly makes clear the importance of multiple technical replicates (i.e., 3 or more ‘runs’) in any bottom-up analysis of any sample. Nonetheless, regarding all these data, it is unclear what value the aggregated data serve when collected using different methods, and only BSA was analyzed.

Figures. Please provide asterisk to denote significance where applicable and describe the test(s) used in the Materials and Methods section.

Please note that there are minor typos throughout, including in the Suppl data.

The Discussion and Conclusion are very short. Please expand on your results and the importance of your findings and how this relates back to proteoforms (as mentioned in introduction). The generalizations presented are largely based only on analysis of a highly purified BSA sample. Such assumptive generalizations are thus lacking in terms of genuine proteomic analyses. It is difficult to support publication without such comparably detailed analyses of at least 2-3 different native samples (i.e., native proteomes), and also a clear demonstration that quantitative comparisons between two similar samples are possible (e.g., in some manner treated vs untreated), without variability of peptide isolation causing quantitative artefacts.

Round 2

Reviewer 2 Report

Fundamentally i must recommend that the publication be rejected due to very low utility. The additional experiments that were performed on complex mixtures, actually demonstrated that the method used is in fact not effective.   The supplementary files reported a total of ~600 identified proteins in yeast and only ~60 in plasma. although extensive peptide characterisation is done on the data that was collected. the methods as reported, is only able to detect 25% of the commonly detected proteome in plasma and only 15% of the described yeast proteome (the literature is extensive in this area).    With this in mind there are significant questions raised regarding the actual action of salt mediated precipitation and whether the lack of proteins was due to inadequate digestion or due to a bias in peptides that are precipitated.    The work needs to be fundamentally revisited and the experimental design needs to be re-thought.    If the authors choose to limit the scope of this paper to purified protein analysis then the method is sufficiently supported. However, the general interest of this work would be very limited and as such unsuitable for publication in Proteomes. 

Author Response

We can appreciate the concerns of the reviewer, as we agree that our MS profiling of complex proteome systems does not compete with state-of-the-art MS investigations of equivalent samples. We note the MS platform, an Orbitrap Velos Pro, is a decade-old instrument, which limits the capacity for in-depth proteome profiling.  The number of peptides reported in this study (>4000 unique identifications) is consistent with the performance of this MS platform, which unfortunately happens to be the most advanced MS system available in the Atlantic region of Canada, and the best we could access.

We must emphasize that the goal of this study was not to present a 'longer list' of identified proteins.  In other words, the MS platform chosen in no way limited the goals of our work. Rather, our study was designed to disclose the following:

1) Is the peptide precipitation protocol broadly applicable, or limited to a certain class of peptides (displaying specific amino acids, or over a set range of properties such as molecular weight).

2) Is the peptide precipitation protocol also applicable to 'real world', complex proteome systems?

The second question was in response to a concern by each of the three reviewers from round 1, and we feel it was an excellent suggestion to strengthen this manuscript.  The revised submission includes two such samples (yeast, plasma).  And the net result of this work was to add to the statistical confidence of our initial conclusions.

While the reviewer is correct that our study reported only a small fraction of the total proteins that are potentially present in the complex systems studied, as our study clearly demonstrates, these 'missing' proteins are not a function of limitations in our sample preparation.

The reviewer first asks whether the lacking proteins are a result of bias in the peptides precipitated. Our manuscript has clearly disclosed that some bias does exist: lower molecular weight peptides partition to a lesser extent into the precipitation pellet for example.  This does not imply that small peptides cannot be recovered - only that the smaller peptides show a statistically measurable drop in recovery.  From line 311, over 20% of identified peptides were below 1000 Da.  Figure 6 compares the molecular weight profile, as well as the hydrophobicity and pI of peptides recovered in the pellet vs supernatant. We observe statistically significant differences, and discuss that certain peptides show more favorable recovery depending on these properties. 

Most significantly, as shown in Figure 5, over 96 % of the MS-identified peptides were recovered in the pellet fraction. Given only minimal differences in their properties, this is clear evidence supporting that the vast majority of peptides are in fact recovered in the pellet fraction.  This complements the fact we observe over 90% yield, reported in Figure 4.

The reviewer also speculates that the poor identification may be a function of poor pepsin digestion efficiency. One needs only to examine the list of peptides identified as evidence this is not the case. Our digestion protocol is based on literature recommendations.  Our group has performed detailed enzyme activity assays, and routinely examines the digestion products by SDS PAGE.  We also attempt to precipitate the sample in 80% acetone + 100 mM NaCl - a protocol which we have previously reported to recover "intact" proteins. The failure to recover sample in appreciable yield under these conditions (Figure 1) is another proof that our digestion was successful.

A far more likely explanation for the missing proteins relates to the limitations of the LC-MS/MS platform employed. This is inconsequential to the study. The reviewer's suggestion that our precipitation protocol is incapable of broad protein recovery is unsupported, since we have clearly demonstrated that the opposite is true.

To ensure that the readers of this manuscript fully appreciate the limitations of the MS platform, we have revised the results and discussion to clearly describe the MS limitations, the goals of the study, and how the results should be interpreted.

While advances in MS instrumentation have undoubtedly contributed to an explosion of growth in the proteomics field, reliable and unbiased sample preparation protocols are essential to the characterization. The peptide precipitation protocol described here is therefore an important contribution with relevance to those interested in low molecular weight peptide/ protein/ proteome profiling.

We thank the reviewer for their time, and their candid remarks, as they have once again provided us an opportunity to strengthen the presentation of this work.

Reviewer 3 Report

No further comments

Author Response

We thank the reviewer for their many helpful suggestions and assistance in clarifying the presentation of our results.